# Estimation of total cardiovascular risk using the 2019 WHO CVD prediction charts and comparison of population-level costs based on alternative drug therapy guidelines: a population-based study of adults in Bangladesh

Jessica Yasmine Islam [1,2] M Mostafa Zaman,[1] Mohammad Moniruzzaman [3] Shawkat Ara Shakoor,[4] A H M Enayet Hossain[4]

For numbered affiliations see end of article.

**Correspondence to**
Dr Jessica Yasmine Islam;
islamjy@email.unc.edu

## ABSTRACT

**Objective** The objective of this study was to estimate the population distribution of 10-year cardiovascular disease (CVD) risk among Bangladeshi adults aged 40 years and above, using the 2019 WHO CVD risk prediction charts. Additionally, we compared the cost of CVD pharmacological treatment based on the total CVD risk (thresholds ≥30%/≥20%) and the single risk factor (hypertension) cut-off levels in the Bangladeshi context.
**Study design** Cross-sectional, population-based study.
**Setting and participants** From 2013 to 2014, we collected data from a nationally representative cross-sectional survey of adults aged ≥40 years from urban and rural areas of Bangladesh (n=6189). We estimated CVD risk using the 2019 WHO CVD risk prediction charts and categorised as very low (<5%), low (5% to <10%), moderate (10% to <20%), high (20% to <30%) and very high risk (≥30%). We estimated drug therapy costs using the lowest price of each drug class available (aspirin, thiazide diuretics, statins and ACE inhibitors). We compared the total cost of drug therapy using the total CVD risk versus single risk factor approach.
**Primary outcome measures** Our primary outcome was 10-year CVD risk categorised as very low (<5%), low (5% to <10%), moderate (10% to <20%), high (20% to <30%) and very high risk (≥30%).
**Results** The majority of adults (85.2%, 95% CI 84.3 to 86.1) have a 10-year CVD risk of less than 10%. The proportion of adults with a 10-year CVD risk of ≥20% was 0.51%. Only one adult was categorised with a 10-year CVD risk of ≥30%. Among adults with CVD risk groups of very low, low and moderate, 17.4%, 27.9% and 41.4% had hypertension (blood pressure (BP) ≥140/90) and 0.1%, 1.7% and 2.9% had severe hypertension (BP ≥160/100), respectively. Using the total CVD risk approach would reduce drug costs per million populations to US$144 540 (risk of ≥20%).
**Conclusion** To reduce healthcare expenditure for the prevention and treatment of CVD, a total risk approach using the 2019 WHO CVD risk prediction charts may lead to cost savings.

### Strengths and limitations of this study

► Using the recently updated 2019 WHO cardiovascular disease (CVD) risk prediction charts, this study provides evidence for incorporating WHO CVD risk prediction charts into CVD management and healthcare guidelines, and may lead to potential cost savings from a societal perspective.

► The 2019 WHO CVD risk prediction charts should be applied to a population who have not experienced a CVD event in the past; however, we were unable to confirm self-reported medical history of participants using medical charts or health records, leading to the potential for measurement error due to recall bias.

► The cost estimates we present are an underestimate of total costs for CVD-related treatment as the focus of this study is on the cost of pharmacological intervention only as the largest contributor to overall direct costs in Bangladesh.

► Although we present the total number of people estimated to require drug treatment using 2014 population data, we were unable to identify population estimates of only those at risk of their first CVD event due to lack of surveillance data.

► The CVD 10-year risk cut-offs were defined using risk prediction models derived from 85 cohorts mostly from high-income countries, as data from large-scale prospective cohort data from most low-income and middle-income countries are limited.

## BACKGROUND

Globally, cardiovascular disease (CVD) is the leading cause of death and disproportionately impacts low/middle-income countries (LMICs), where over 75% of CVD-related deaths occur.[1] People living in LMICs are at high risk of developing CVDs due to the absence of integrated primary care for early

detection and prevention of CVD-related risk factors, such as high cholesterol, high blood pressure (BP) and smoking. Limited access to primary care and the growing burden of CVDs are significant causes of poverty in LMICs and hinder the macroeconomic development of many countries.[2] LMICs are estimated to experience cumulative economic losses exceeding US$7 trillion over the next 15 years due to morbidity and mortality caused by non-communicable diseases (NCDs), including CVD.[3] As such, the significance of the CVD epidemic has gained increasing international recognition over the past decade, leading to the development of several international guidelines for CVD control and prevention.[4]

In 2007, WHO published pocket guidelines, including CVD risk prediction charts, designed for healthcare workers in LMICs to guide patient 10-year risk stratification for heart attack or stroke.[5] There are two possible strategies suitable for a low-resource setting to assess the risk of a cardiovascular event and identify those at high risk of a fatal CVD outcome: (1) use a single risk factor management strategy, which focuses on one condition at a time, such as hypertension; or, (2) use a more holistic approach considering several risk factors such as age, tobacco use, gender, diabetes diagnosis, body mass index (BMI), BP and blood cholesterol when measured. Through the total risk approach, pocket guidelines help to identify high-risk patients that are in imminent danger of a heart attack or stroke for timely pharmacological treatment or surgical interventions. Additionally, applying the total risk approach via WHO prediction charts in a nationally representative sample provides an opportunity to estimate and monitor the population-level distribution of CVD risk to ultimately inform CVD treatment policy recommendations.[6 7]

Currently, Bangladesh has not incorporated clinical guidelines for screening or treatment of risk factors based on absolute CVD risk scores and estimation of the population distribution of CVD risk over time. Data are needed to support the implementation of WHO prediction charts as clinical guidelines for Bangladesh, a resource-limited setting, to demonstrate the potential cost savings and benefit of WHO recommendation on CVD prevention. Recently in 2019, WHO updated the CVD risk charts based on newly validated risk prediction models to estimate CVD risk in 21 Global Burden of Disease (GBD) regions.[8] The newly developed risk prediction models have been calibrated using data from the GBD study to include estimates from LMICs. To our knowledge, here we present the first analysis to apply the updated CVD risk charts among a cohort of Bangladeshi adults. Prior studies conducted in Bangladesh have estimated CVD risk among adults residing in rural areas only and have not included a nationally representative population.[9–11] Additionally, no prior studies have estimated the potential costs of pharmacological treatment for CVD in Bangladesh using either the single risk factor or total CVD risk approach, as done previously in other settings.[7] Our objective was to assess the distribution of absolute CVD risk among a nationally representative sample of Bangladeshi adults using the 2019 WHO CVD risk prediction charts recommended for WHO South Asian Region (Bangladesh, Bhutan, India, Nepal and Pakistan). We also compared the costs of drug treatments for CVD prevention using the total cardiovascular risk thresholds at ≥20% and with single risk factor cut-off levels (BP ≥140/90 mm Hg).

## METHODS

### Study design and setting

Data were analysed from a population-based cross-sectional study conducted from September to December 2013 to assess the burden of blindness and low vision among adults in Bangladesh. The target population of this survey included men and women residing in Bangladesh over the age of 40 years. The exclusion criteria included tourists and the institutionalised, such as residents of a military base, hospital, prisons, nursing homes and other such institutions. We provided participants with detailed information regarding the study objectives and procedures using a printed handout prepared in Bengali. Written consent was obtained from participants through signature or, if not possible through thumbprint.

### Sampling frame

We adopted a multistage, geographically clustered, probability-based sampling approach to obtain a nationally representative sample of Bangladesh, as previously described and outlined per WHO STEPwise approach to surveillance (STEPS).[12–15] Population statistics were obtained using the 2011 national census conducted by the Bangladesh Bureau of Statistics to create the sample frame.[16] The sampling frame included 64 407 primary sampling units (PSUs) covering all 7 divisions of the country. We randomly selected 72 PSUs (25 urban and 47 rural) from 7 divisions, with the probability of selection proportional to the population size of each division. In each PSU, we selected 100 consecutive households as the secondary sampling unit.

For each household, a trained field data collector approached the head of the household or the family member most knowledgeable of the residents to screen for eligible participants. The screening respondent was asked to describe the composition of household residents, which was defined as those who considered the home to be their primary place of residence as of the night before. A list was composed and ordered from the youngest to the oldest age in years starting from 40 years. Using the list of eligible residents, we used the Kish table approach to randomly select one participant from each home. The selected participant was asked to come to a nearby health centre the next day to administer the survey by trained study interviewers and undergo a medical examination by the study physician. Based on the medical review, participants were followed up with by the providers at the health centre for treatment.

## Patient and public involvement

There was no patient or public involvement in the implementation of this study or interpretation of analytic results.

## Data collection

To ensure effective and uniform data collection, field interviewers underwent a 7-day training on the interview methodology by the study ophthalmologists and epidemiologists. The training included an in-depth review of the survey content and protocol for completing the demographic questionnaire (a modified WHO/PBL Version III). Each member of the data collection team was provided a detailed survey protocol manual outlining the survey activities, the questionnaire interview and information about the duties and responsibilities of all survey personnel.

Demographic data were collected, including age, sex, marital status, educational level and occupation, using a structured questionnaire survey. Data regarding tobacco use, health history and treatment history were also collected. Participants were asked if they smoked (eg, cigarette, hookah, pipe) or if they used smokeless tobacco (eg, chewing tobacco, jorda) to assess the history of tobacco use. Each participant provided medical history for a prior diagnosis of high BP or hypertension, diabetes, renal disease or any CVDs by a healthcare provider. Medication history was obtained including medication for high BP, diabetes, malaria, steroids, tuberculosis and among women, history of oral contraception. The questionnaire was translated from English to Bengali, adapted, and validated before data collection.

Physical measurements, including height, weight and BP, were collected. Trained field interviewers measured BP using an appropriately calibrated aneroid sphygmomanometer with appropriately sized arm cuffs. BP measurements were consistently taken on the right arm at heart level and elbow assisted while the participant was seated. The initial measurement was performed after 5 min of rest. After 2 min, the second measurement was taken. The mean of these two BP readings was used as the final BP for each participant. To measure blood glucose levels, we obtained random blood glucose samples.[17] Capillary blood samples were consistently taken using the right arm and index finger with a glucometer (Accuchek Advantage, Roche Diagnostics Division, Grenzacherstrasse, Switzerland).

## Estimation of 10-year CVD risk

We estimated 10-year CVD risk using the 2019 WHO CVD risk prediction charts.[8 18] The prediction charts provide the 10-year risk of a fatal or non-fatal major cardiovascular event, such as myocardial infarction or stroke, based on age, sex, BP, BMI, smoking status, total blood cholesterol and the presence or absence of diabetes mellitus for 14 WHO epidemiological subregions. For each region, two sets of charts have been developed based on the availability of laboratory-based results. As total cholesterol was not measured in our cohort, we used WHO CVD risk non-laboratory-based charts developed for South Asia (including Bangladesh, Bhutan, India, Nepal and Pakistan). The non-laboratory-based risk charts do not account for diabetes diagnosis or total cholesterol levels.

The prediction chart grades CVD risk using the following categories: age (1: 40–44 years; 2: 45–49 years; 3: 50–54 years; 4: 55–59 years; 5: 60–64 years; 6: 64–69 years; 7: 70–74 years), sex (men and women), smoking (smoker or non-smoker), systolic BP (SBP; <120 mm Hg, 120–139, 140–159, 160 to <180 and ≥180), and BMI (<20, 20–24, 25–29, 30–35 and ≥35). The risk categories for 10-year combined acute myocardial infarction and stroke (fatal and non-fatal) are as follows:<5%, 5% to <10%, 10% to <20%, 20% to <30% and ≥30%.

Observations with missing values were dropped from the analysis. We did not anticipate any bias from the complete-case analysis approach as the number of missing observations for key variables was less than 2%: smoking status, n=18 or 0.3% missing values; SBP, n=22 or 0.4% missing values; diastolic BP (DBP), n=25 or 0.4% missing values; and BMI, n=30 or 0.5%,

## Data analysis

We present sociodemographic variables using mean (SD) or the median (IQR) for continuous variables and proportion for categorical variables. We conducted bivariate analyses by sex and age group. We used the $\chi^2$ test to assess for any significant differences in CVD risk distribution across sex. For estimating the cost of medicines per million per year (population aged 40 years or older), we used the lowest price of each drug class available in the market (generic preparation of aspirin, thiazide diuretics, statins and ACE inhibitors). Online supplementary appendix table 1 includes further details regarding the specific costs of common drugs used to treat CVD in Bangladesh.

T0 calculate costs, we included the following categories: (1) people with high cardiovascular risk (≥20% and BP ≥160/100), who are recommended for pharmacological intervention using four different types of drugs for treatment[5 7]; and (2) people with BP ≥140/90, who are recommended antihypertensive treatment. To calculate the estimated annual total cost of CVD medication treatment per million populations (aged 40 years or older), we multiplied the percentage of the population at risk and the price of medicine in Bangladesh. We included an estimate of the total number of people estimated to require drug treatment as follows: we multiplied the prevalence of the population requiring medication based on each approach by the number of people in the general population in 2013[19] stratified by gender and age group.

## RESULTS
### Demographic characteristics
The mean age of included participants was 52.9 years (men: 53.5 years, women: 52.5 years; table 1). The

**Table 1** Background characteristics of Bangladeshi adult participants (n=6189)

| Characteristics | Total (n=6189) | | Men (n=2824) | | Women (n=3365) | |
|---|---|---|---|---|---|---|
| | Mean (SD) | n (%) | Mean (SD) | n (%) | Mean (SD) | n (%) |
| Age (years) | 52.9 (9.9) | | 53.5 (10.1) | | 52.5 (9.8) | |
| Education (years) | 3.1 (4.2) | | 4.3 (4.8) | | 2.1 (3.3) | |
| Area of residence | | | | | | |
| Urban | | 1873 (30.2) | | 808 (28.6) | | 1065 (31.6) |
| Rural | | 4316 (69.8) | | 2016 (71.4) | | 2300 (68.4) |
| Occupation | | | | | | |
| Professional employment* | | 1007 (16.3) | | 880 (31.3) | | 127 (3.8) |
| Industrial worker/day labourer | | 1600 (25.9) | | 1440 (51.2) | | 160 (4.8) |
| Housework | | 2697 (43.7) | | – | | 2697 (80.3) |
| Unemployed/retired | | 735 (11.9) | | 386 (13.7) | | 349 (10.4) |
| Other† | | 131 (2.1) | | 106 (3.8) | | 25 (0.7) |
| Smoking tobacco use‡ | | | | | | |
| Current user | | 1493 (24.2) | | 1432 (50.7) | | 61 (1.8) |
| Ever user | | 2261 (36.5) | | 2150 (76.1) | | 111 (3.3) |
| Smokeless tobacco use§ | | | | | | |
| Current user | | 3078 (49.7) | | 1208 (42.8) | | 1870 (55.6) |
| Ever user | | 3469 (56.1) | | 1433 (50.7) | | 2046 (60.5) |
| Alcohol use in the last 30 days | | | | | | |
| Yes | | 75 (1.2) | | 68 (2.4) | | 7 (0.2) |
| Body mass index¶ | 21.9 (4.1) | | 21.4 (3.7) | | 22.3 (4.4) | |
| Waist circumference** (cm) | 82.4 (13.3) | | 81.8 (10.5) | | 82.8 (15.2) | |
| Blood pressure | | | | | | |
| Systolic blood pressure†† (mm Hg) | 119.7 (15.2) | | 119.8 (14.9) | | 119.7 (15.5) | |
| Diastolic blood pressure‡‡ (mm Hg) | 80.3 (9.5) | | 80.2 (9.4) | | 80.3 (9.7) | |
| Blood glucose levels§§ (mmol/L) | 6.9 (3.0) | | 6.9 (3.0) | | 6.9 (2.9) | |

*Professional occupation includes: government employee, private company employee, businessman.
†Other occupation includes: shopkeeper, weavers, driver, student, beggar, cook, carpenter and tailor.
‡Excluding smokeless tobacco|missing values, n=18; men, n=11; women, n=7.
§Smokeless tobacco use includes jorda, white leaf (shaada pata), gul and so on.
¶Body mass index calculated by weight in kilogram divided by height in metre squared
**Missing values, n = 58; Male, n = 25; Female, n = 33
†† Missing values, n = 22; Male, n = 13; Female, n = 9
‡‡Missing values, n = 25; Male, n = 15; Female, n = 10
§§Missing values, n = 84; Male, n = 36; Female, n = 48
CM, Centimeters; mmHg, Millimeter of mercury; SD, Standard Deviation.

average level of educational attainment was 3.1 years of education; women were generally less educated than men (2.1 years vs 4.3 years, respectively). The majority (80%) of women were housewives, and among men, the most common occupation was an industrial worker or day labourer (51.2%). Overall, over one-third of participants ever used smoking tobacco, and over half ever used smokeless tobacco. Few participants drank alcohol in the past 30 days (1.2%). The mean BMI was 21.9 kg/m$^2$, and the mean waist circumference was 82.4 cm.

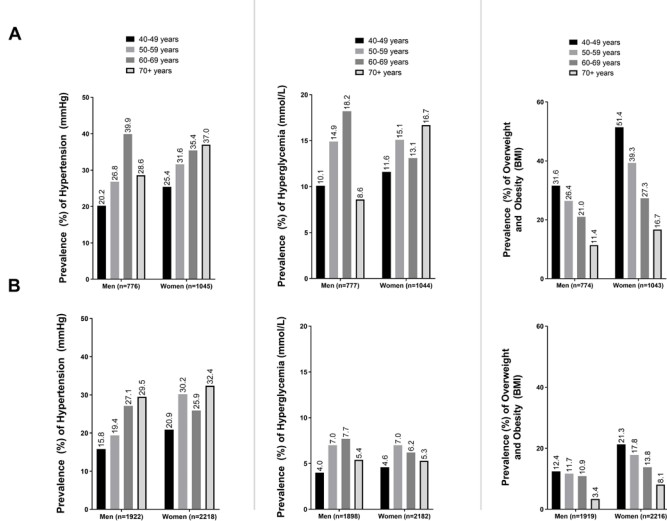

A

B

**Figure 1** Prevalence of hypertension, hyperglycaemic and overweight and obesity among Bangladeshi adults aged 40 years and above by area of residence and age group. (A) Urban populations. (B) Rural populations. BMI, body mass index.

Overall, the prevalence of hypertension (defined as BP ≥140/90) increased with age among men and women. Additionally, women had a higher prevalence of hypertension among nearly all age groups in both urban and rural areas. The prevalence of hyperglycaemic (glucose ≥11.1 mmol/L) was higher among urban adults compared with rural across all age groups. The highest prevalence of hyperglycaemic was observed among urban men aged 60–69 years at 18.2%. Finally, the prevalence of overweight and obesity was higher among urban residents than rural residents. The largest prevalence of overweight and obesity was observed among urban women, with prevalence as high as 51.4% among women aged 40–49 years (figure 1).

### Distribution of cardiovascular risk

We summarised the distribution of CVD risk in the population overall and stratified by sex in table 2. Eighty-five per cent of participants had a low (<10%) 10-year CVD risk, and this proportion was significantly different across

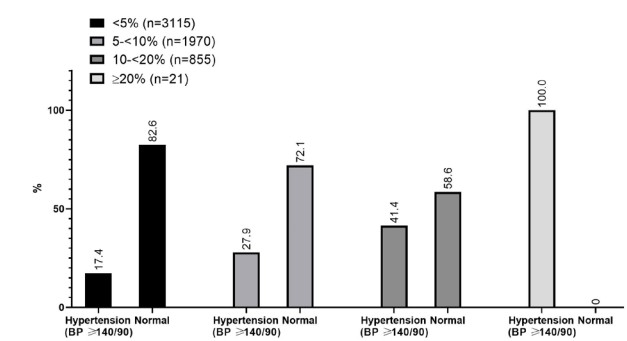

A

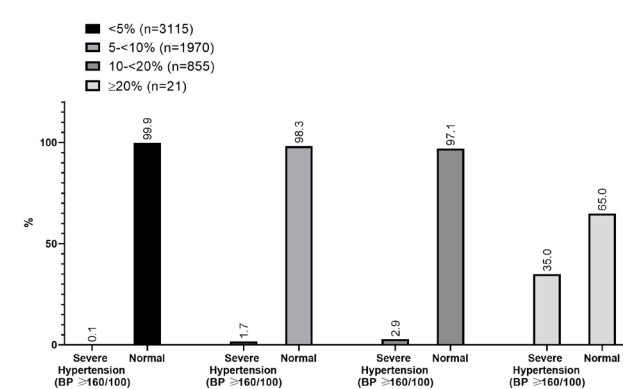

B

**Figure 2** Proportion of adults with hypertension or severe hypertension by cardiovascular disease (CVD) risk group. (A) Per cent of adults with hypertension by CVD risk group. (B) Per cent of adults severe with hypertension by CVD risk group. BP, blood pressure.

sex (p<0.001). Over half (63.7%) of women had a very low (<5%) cardiovascular risk. Almost all (99.5%) of the study population were categorised as having cardiovascular risk <20%. A higher proportion of men (1.0%) were categorised as high risk than women (0.1%; p =<0.001). Overall, only one male participant was categorised as very high risk or with a CVD risk of ≥30%.

We summarised the prevalence of adults with hypertension by CVD risk group (figure 2). Among those with 10% to <20% CVD risk, we observed a high proportion of hypertension (41.4%). In the high-risk group (≥20%),

**Table 2** Ten-year risk of combined myocardial infarction and stroke (fatal and non-fatal) by gender, using the 2019 WHO cardiovascular disease risk non-laboratory-based charts for South Asia (n=5977)

| | Total (n=5977) | | | Men (n=2708) | | | Women (n=3269) | | | P value* |
|---|---|---|---|---|---|---|---|---|---|---|
| | n | % | CI | n | % | CI | n | % | CI | |
| Very low risk (<5%) | 3115 | 52.1 | 50.8 to 53.4 | 1034 | 38.2 | 36.6 to 40.0 | 2081 | 63.7 | 62.0 to 65.3 | <0.001 |
| Low risk (5%–10%) | 1972 | 33.0 | 31.8 to 34.2 | 1047 | 38.7 | 36.8 to 40.5 | 925 | 28.3 | 26.8 to 30.0 | <0.001 |
| Moderate risk (10% to <20%) | 860 | 14.4 | 13.5 to 15.3 | 600 | 22.2 | 20.6 to 23.8 | 260 | 7.8 | 7.0 to 8.9 | <0.001 |
| High risk (20% to <30%) | 29 | 0.5 | 0.3 to 0.7 | 26 | 1.0 | 0.6 to 1.4 | 3 | 0.1 | 0.0 to 0.2 | <0.001 |
| Very high risk (≥30%) | 1 | 0.0 | 0.0 to 0.01 | 1 | 0.0 | 0.0 to 0.1 | 0 | 0.0 | 0.0 to 0.0 | 0.272 |

*P value based on $\chi^2$ test.

**Table 3** Estimation of percentage of population requiring drug treatment based on total risk approach in comparison to single risk factor approaches (n=5977)

|  | Cardiovascular (CV) risk ≥20% | | Single risk factor approach |
| --- | --- | --- | --- |
|  | CV risk ≥20% alone, % | CV risk ≥20% + BP ≥160/100 | BP ≥140/90 (SBP ≥140 + isolated raised DBP), % |
| Men | 1.0 | 2.1 | 22.1 |
| Women | 0.1 | 1.6 | 26.7 |
| Total | 0.5 | 1.8 | 24.6 |

BP, blood pressure; DBP, diastolic blood pressure; SBP, systolic blood pressure.

100% had hypertension. Additionally, among those with ≥20% CVD risk, we observed that 35% had severe hypertension (BP ≥160/100).

### Cost of drug treatment

We were unable to compare the costs of drug treatment at two cardiovascular risk thresholds (30%–20%) due to only one male adult with a CVD risk at ≥30%. We observed a low proportion of adults with CVD risk ≥20% at 0.5%. When we included BP≥160/100 measurements, the number of people requiring treatment more than tripled from 0.5% to 1.8% (table 3). Conversely, if a single risk factor approach was applied, and all those with hypertension (a persistent SBP ≥140 and/or DBP≥90) were treated, 24.6% of the sample would require drug treatment, specifically antihypertensive; more than 20 times the proportion identified when using the total cardiovascular risk approach alone.

### Comparison of cost by approach

Next, we compared the estimated annual cost of medicines per million populations for implementing the total risk approach versus the single risk factor approach. Table 4 shows the estimated number of people aged 40 years or older requiring drug treatment stratified by age group and gender. The estimates showed that if the single risk factor approach is applied in Bangladesh with its percentage of the population at risk and the lowest price of medicine in the country, the cost per million populations (aged 40 years or older) of treating those with BP ≥140/90 would be US$7 111 368; if the absolute risk approach were applied, the cost of treating those with a 10-year risk of CVD ≥20% per million populations (aged 40 years or older) would be US$144 540, almost 50 times less (figure 3). The cost estimation was based on the percentage of the population at different levels of risk and the differences in the price of generic medicines. For this analysis, we focused on the cost of pharmacological treatment as it is the most critical contributor to the overall direct costs of CVD treatment in Bangladesh. We assumed that other costs of CVD treatment and prevention service delivery, such as health facilities and wages of health workers, are similar for both approaches of service delivery.[7]

### DISCUSSION

Using this nationally representative survey of Bangladeshi adults aged 40 years and above, we found that the majority of adults (97%) were at a very low, low or moderate 10-year risk of myocardial infarction and stroke. The proportion of adults requiring drug treatment rose from 1.0% to 2.1% when the threshold for pharmacological intervention was changed from ≥20% alone to ≥20% plus BP of 160/100, respectively, which was lower the proportion than the single risk factor approach (24.6%). Our data demonstrate that using a single risk factor approach to manage individual cardiovascular risk factors is costlier (US$7 111 368 per million population) than using the total risk approach (CVD risk ≥20, US$144 540 per million population), as a more substantial proportion of adults will need drug treatment. Findings from this analysis support the implementation of clinical guidelines using CVD risk scores calculated using WHO CVD risk prediction charts to appropriately identify patients at the highest risk of CVD development over 10 years in Bangladesh.

In our study using nationally representative data, we found that the 10-year risk of CVD was low (<10%) among the vast majority of adults (85.1%). Additionally, only 0.5% of adults were at high risk (≥20%) of a CVD event within the next 10 years. In this analysis, we applied the 2019 CVD risk charts for South Asia, which are newly developed and now incorporate BMI as part of the prediction chart algorithm. Our results are comparable to regional data presented in the 2019 Lancet publication by WHO CVD Risk Chart Working Group, which showed that 0% of women from Bhutan, Sri Lanka and Nepal had a CVD risk level above 20%. Similarly, 0% of men from Bhutan, and only <2% of men from both Sri Lanka and Nepal were categorised with a risk level above 20%.[8] These data demonstrate a lower prevalence of CVD risk ≥20% than prior reports from South Asia, which used the original risk prediction charts published in 2007. For example, prior data from Nepal[20] and Sri Lanka[21] showed that 4.3% and 8.2% of adults, respectively, were categorised with a high (≥20%) 10-year risk of a CVD event. Further, analyses from a rural area of South India revealed that 17% of participants had moderate-to-high risk (10%–>20%) of cardiovascular events per the 2007

**Table 4** Population projections and comparison of medicine costs (US$) for implementing total risk approach versus single risk factor approach in Bangladesh

| | Total no of people in the general population (in thousands)* | Percentage of population aged ≥40 years requiring medication (%) | Total no of people estimated to require drug treatment (in thousands) | No of people per million population (aged 40 years and older) | Aspirin (US$) | Enalapril (US$) | Hydrochlorothiazide (US$) | Simvastatin (US$) | Total (US$) |
|---|---|---|---|---|---|---|---|---|---|
| **Total risk approach CV risk ≥20% alone** | | | | | | | | | |
| **Age group** | | | | | | | | | |
| **Men** | | | | | | | | | |
| 40–49 | 9210 | 0.4 | 3660 | 4000 | 6570.0 | 34748.0 | 12118.0 | 62196.0 | 115632.0 |
| 50–59 | 6303 | 0.1 | 596 | 1000 | 1642.5 | 8687.0 | 3029.5 | 15549.0 | 28908.0 |
| 60–69 | 3730 | 1.5 | 4733 | 15000 | 24637.5 | 130305.0 | 45442.5 | 233235.0 | 433620.0 |
| ≥70 | 1881 | 7.0 | 17507 | 70000 | 114975.0 | 608090.0 | 212065.0 | 1088430.0 | 2023560.0 |
| Total | 21124 | 1.0 | 20769 | 10000 | 16425.0 | 86870.0 | 30295.0 | 155490.0 | 289080.0 |
| **Women** | | | | | | | | | |
| 40–49 | 9087 | 0.0 | 0 | 0 | 0.0 | 0.0 | 0.0 | 0.0 | 0.0 |
| 50–59 | 5662 | 0.0 | 0 | 0 | 0.0 | 0.0 | 0.0 | 0.0 | 0.0 |
| 60–69 | 3257 | 0.0 | 0 | 0 | 0.0 | 0.0 | 0.0 | 0.0 | 0.0 |
| ≥70 | 1638 | 1.6 | 3965 | 16000 | 26280.0 | 138992.0 | 48472.0 | 248784.0 | 462528.0 |
| Total | 19644 | 0.1 | 1958 | 1000 | 1642.5 | 8687.0 | 3029.5 | 15549.0 | 28908.0 |
| **All** | | | | | | | | | |
| 40–49 | 18296 | 0.2 | 3605 | 2000 | 3285.0 | 17374.0 | 6059.0 | 31098.0 | 57816.0 |
| 50–59 | 11965 | 0.1 | 1134 | 1000 | 1642.5 | 8687.0 | 3029.5 | 15549.0 | 28908.0 |
| 60–69 | 6989 | 0.7 | 4206 | 7000 | 11497.5 | 60809.0 | 21206.5 | 108843.0 | 202356.0 |
| ≥70 | 3518 | 4.3 | 21410 | 43000 | 70627.5 | 373541.0 | 130268.5 | 668607.0 | 1243044.0 |
| Total | 40768 | 0.5 | 20175 | 5000 | 8212.5 | 43435.0 | 15147.5 | 77745.0 | 144540.0 |
| **Single risk factor approach: BP ≥140/90 (SBP ≥140 + isolated raised DBP), %** | | | | | | | | | |
| **Age group** | | | | | | | | | |
| **Men** | | | | | | | | | |
| 40–49 | 9210 | 17.1 | 156482 | 171000 | 280867.5 | 1485477.0 | 518044.5 | 2658879.0 | 4943268.0 |
| 50–59 | 6303 | 21.7 | 129375 | 217000 | 356422.5 | 1885079.0 | 657401.5 | 3374133.0 | 6273036.0 |
| 60–69 | 3730 | 30.5 | 96228 | 305000 | 500962.5 | 2649535.0 | 923997.5 | 4742445.0 | 8816940.0 |
| ≥70 | 1881 | 29.4 | 73529 | 294000 | 482895.0 | 2553978.0 | 890673.0 | 4571406.0 | 8498952.0 |

Continued

**Table 4** Continued

Total risk approach CV risk ≥20% alone

| | Total no of people in the general population (in thousands)* | Percentage of population aged ≥40 years requiring medication (%) | Total no of people estimated to require drug treatment (in thousands) | No of people per million population (aged 40 years and older) | Estimated annual total cost of CVD medication treatment per million population† | | | | |
|---|---|---|---|---|---|---|---|---|---|
| | | | | | Aspirin (US$) | Enalapril (US$) | Hydrochlorothiazide (US$) | Simvastatin (US$) | Total (US$) |
| Total | 21 124 | 22.1 | 458 995 | 221 000 | 362 992.5 | 1 919 827.0 | 669 519.5 | 3 436 329.0 | 6 388 668.0 |
| **Women** | | | | | | | | | |
| 40–49 | 9087 | 22.5 | 199 643 | 225 000 | 369 562.5 | 1 954 575.0 | 681 637.5 | 3 498 525.0 | 6 504 300.0 |
| 50–59 | 5662 | 30.6 | 164 475 | 306 000 | 502 605.0 | 2 658 222.0 | 927 027.0 | 4 757 994.0 | 8 845 848.0 |
| 60–69 | 3257 | 28.7 | 81 881 | 287 000 | 471 397.5 | 2 493 169.0 | 869 466.5 | 4 462 563.0 | 8 296 596.0 |
| ≥70 | 1638 | 33.7 | 83 509 | 337 000 | 553 522.5 | 2 927 519.0 | 1 020 941.5 | 5 240 013.0 | 9 741 996.0 |
| Total | 19 644 | 26.7 | 522 759 | 267 000 | 438 547.5 | 2 319 429.0 | 808 876.5 | 4 151 583.0 | 7 718 436.0 |
| **All** | | | | | | | | | |
| 40–49 | 18 296 | 20.2 | 364 085 | 202 000 | 331 785.0 | 1 754 774.0 | 611 959.0 | 3 140 898.0 | 5 839 416.0 |
| 50–59 | 11 965 | 26.3 | 298 189 | 263 000 | 431 977.5 | 2 284 681.0 | 796 758.5 | 4 089 387.0 | 7 602 804.0 |
| 60–69 | 6989 | 29.5 | 177 236 | 295 000 | 484 537.5 | 2 562 665.0 | 893 702.5 | 4 586 955.0 | 8 527 860.0 |
| ≥70 | 3518 | 31.6 | 157 336 | 316 000 | 519 030.0 | 2 745 092.0 | 957 322.0 | 4 913 484.0 | 9 134 928.0 |
| Total | 40 768 | 24.6 | 992 585 | 246 000 | 404 055.0 | 2 137 002.0 | 745 257.0 | 3 825 054.0 | 7 111 368.0 |

*Source: Population Projections (using 2011 Census data) Bangladesh Bureau of Statistics, Statistics and Informatics Division, Ministry of Planning
†Price for 100 tablets in US$: Aspirin (0.45), Enalapril (2.38), Hydrochlorothiazide (2.38), Simvastatine (0.83), Simvastatine (4.29); one tablet taken per da
BP, blood pressure; CV, cardiovascular; CVD, cardiovascular disease; DBP, diastolic blood pressure; SBP, systolic blood pressure.

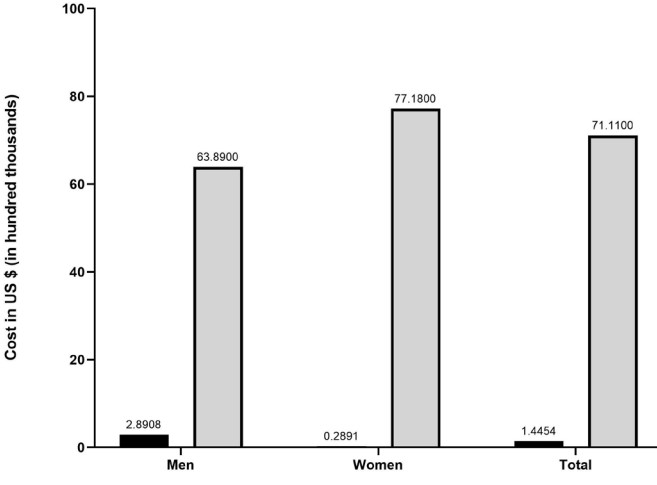

■ Estimated annual cost of CVD treatment using the total risk approach ≥20% alone

▢ Estimated annual cost of CVD treatment using the single risk factor approach

**Figure 3** Annual costs of pharmacological treatment for cardiovascular disease (CVD) by sex and risk stratification approach.

WHO prediction charts.[22] Finally, data collected in 2010 from Pakistan showed that 10% of adults were categorised with ≥20% CVD risk, with 2.9% as high as ≥40%.[7] When using the 2019 WHO prediction charts on the population level to measure and monitor trends in total CVD risk in recent years, policy-makers in LMICs should interpret the trends with caution, and assess changes in trends of CVD risk over time using both the 2007 and 2019 WHO risk prediction charts for comparison.

WHO CVD risk charts were developed for use in LMICs and are now more suitable for use in these settings due to the inclusion of data from low-resource regions in the risk prediction model development. While we present the first country-specific analysis using the 2019 risk charts, several prior studies in LMICs have been conducted using the 2007 risk charts. In other countries in Asia, we observe the prevalence of 'high CVD risk' (≥20%) of 6.0%, 2.3% and 1.3% in Mongolia, Malaysia and Cambodia, respectively.[23] Mendis *et al* reported the 10-year CVD risk of seven countries and the majority of these countries reported low CVD risk among its adult populations (China (96.1%) and Sri Lanka (94.9%), (Iran (93.9%), Cuba (89.7%), Nigeria (86.0%), Georgia (83.1%) and Pakistan (79.2%)).[7] Studies conducted in urban areas of LMICs show varying prevalence of high CVD risk (≥20%): for example, one study from Malaysia shows 20.5% of adults were high risk of CVD,[24] whereas studies from urban Kenya[25] and Sri Lanka[21] report less than 10% of their population had high CVD risk. Specifically in Bangladesh, using the 2007 prediction charts, three prior studies have reported absolute CVD risk among the adult population.[10 26 27] Fatema *et al* found that 10% of rural Bangladeshi adults were at high risk (≥20%) of a CVD event within the next 10 years, and half of these adults fell in the very high-risk category (≥30%). In another rural

Bangladeshi population, the proportion of participants at a high risk (≥20%) of a CVD event was 2.1%.[26] Finally, in an urban Bangladeshi population of 150 adults, 3.4% had high CVD risk (≥20%), which is lower than expected as the population was urban.[27] No other studies in Bangladesh have been conducted to assess the 10-year risk of CVD using WHO CVD risk prediction charts. We present novel data using the 2019 charts and a nationally representative sample, which may be generalisable to the population of Bangladesh. Data we present may be used to inform policy-makers decisions on clinical guidelines and resource allocation for treatment of CVDs in Bangladesh.

Similar to prior analyses conducted using data from eight LMICs,[7] our results demonstrate that in the Bangladeshi context using a single risk factor approach to evaluate the risk of CVD-related mortality would cost more than implementing the total risk approach due to higher drug costs. In Bangladesh, about 60% of out-of-pocket costs patients face go towards drugs directly bought from pharmacies, diagnostics and informal providers.[28] Currently, Bangladesh does not offer universal health coverage or affordable health insurance plans. The cost of treatment for CVD frequently leads to catastrophic health expenditure and impoverishment; the proportion of catastrophic spending for treatment is highest among those from the lowest quintiles of wealth (14%) compared with those with high wealth and high socioeconomic status (6.6%).[29] As such, implementing WHO CVD risk prediction charts may be beneficial to patients in Bangladesh as only those at the highest risk of future CVD would be recommended for treatment.

In addition to benefits to the patient, the total CVD risk approach would also be beneficial to Bangladesh's healthcare system by improving NCD preventive service delivery and the use of guidelines for adequate care. Currently, Bangladesh is categorised by the World Bank as a lower middle-income country with emerging health challenges as the burden of NCDs continues to grow. In 2015, an estimated 67% of all deaths in Bangladesh were due to NCDs and the risk of premature death from chronic disease among adults aged 30–70 years was 22%.[30] Indeed, CVDs and circulatory diseases are the leading causes of mortality and morbidity in Bangladesh. Despite this substantial burden, preventive services for CVDs in Bangladesh are limited. In 2014, an estimated 16% of healthcare facilities across the country (ie, hospitals, community clinics) had the resources to diagnose, prescribe treatment for and manage patients with CVDs.[31] District hospitals (95%), Upazila health complexes (81%) and private hospitals (77%) were more likely to provide services for CVDs than other facilities. Only 10% of community clinics and maternal and child welfare centres, and 17% of union level facilities, which are the most accessible providers in rural areas, provided any cardiovascular services, and the services at these facilities were limited to the measurement of BP or referrals.[31] Among facilities with the capacity to offer services for CVD management, about only 20% used established guidelines for hypertension treatment

and less than one-third had essential CVD medicines readily available on-site for patients.[31] By integrating WHO risk prediction charts into the national guidelines for management of hypertension and CVD prevention in Bangladesh, the proportion of facilities using established guidelines may increase as the charts are easy to implement, interpret and access. Additionally, since only one-third of facilities have essential CVD medicines readily available, distributing pharmacological treatment to those at highest risk of premature mortality due to CVD will be crucial.

Although the implementation of a total risk approach may lead to cost savings, there are limitations to implementing the 2019 CVD risk prediction charts. When compared with the single risk factor approach, WHO charts categorise fewer individuals as high risk and may delay the receipt of necessary life-saving treatment. For example, the prediction charts may underestimate CVD risk in certain categories of people such as those with persistent raised BP ≥160/100 mm Hg, blood cholesterol ≥8 mmol/L, or those suffering from diabetes with renal disease.[5] Patients who may fall in these categories should be recommended for intensive lifestyle interventions and appropriate drug therapy; however, the CVD prediction charts will erroneously deny treatment to these potentially high-risk adults. In fact, the risk models used to develop the 2019 CVD risk charts may have underestimated CVD risk due to limitations in the population data used to estimate incidences: Data used to develop the predictions models likely included people already on CVD prevention therapies, such as statins, which have led to an underestimate in CVD risk.[8] In our study, we underscore the potential for underestimation of CVD risk by comparing the proportion of adults categorised as high risk (≥20% CVD risk) to those who would be diagnosed with hypertension (BP ≥140/60) and severe hypertension (BP ≥160/100). Additionally, we provided a graphical summary of common risk factors of CVD, including hypertension, hyperglycaemia, and overweight and obesity. Despite our very low proportion of adults who would be recommended for treatment based on the risk prediction charts, we observed a high prevalence of these risk factors particularly in urban populations.

Limitations of this analytical approach should be considered when interpreting our results. The CVD 10-year risk cut-offs were defined using risk prediction models derived from 85 cohorts mostly from high-income countries, as data from large-scale prospective cohort data from most LMICs are unavailable. Data were used from the GBD project to recalibrate the models to be representative of LMICs; however, the GBD data do not have country-specific disease risk estimates. As such, the estimation used from each region's chart will most likely apply to the largest country within each region, or from the country where most of the data originated. The risk prediction charts provide approximate estimates of CVD risk in people who do not have established coronary heart disease, stroke or other atherosclerotic diseases. Although

we included simvastatin in our pharmacological cost analysis, we were unable to measure total cholesterol or confirm the medical history of participants using medical charts and relied on self-report, leading to the potential for measurement error and recall bias. Additionally, we were unable to categorise participants as diabetic as we did not obtain fasting blood glucose and were only able to categorise adults as hyperglycaemic in our descriptive analysis as we measured random blood glucose. Further, our data were collected in 2013 and may be outdated as population growth in older age groups has been observed in recent years. Our analyses should be replicated using more recent data and future research studies should include the measurement of total cholesterol. Finally, our cost estimates were based on the prevalence of each risk approach in our study sample. Although we present the total number of people estimated to require drug treatment using 2013 population data, estimates of only those at risk of their first CVD event were unavailable due to lack of surveillance data. Nevertheless, our data are valuable as the first analysis to apply the 2019 WHO CVD risk prediction charts to a cohort of adults in Bangladesh. Additionally, we provide data on the comparative cost difference of each approach to underscore the potential cost savings in implementing the total risk approach in Bangladesh. Cost data presented in this analysis may be used in future cost-effectiveness analyses to compare the total risk and single risk factor approach when considering all costs from a societal perspective to inform health policy in Bangladesh.

WHO has outlined global targets in the Global Monitoring Framework for the control of NCDs in LMICs, which prioritises an 80% of availability of affordable basic technologies and essential medicines necessary to treat significant NCDs, including CVDs in both rural and urban areas of the country. Limited CVD treatment access and weak healthcare infrastructure in Bangladesh are a significant public health concern. As public financing for healthcare is limited in Bangladesh (~1% of gross domestic product), public health policies on CVD drug treatment guidelines based on cost estimates, such as out-of-pocket costs, are necessary for effective CVD control. Effective policies should address the potential for overtreatment, which comes at a high cost to both the healthcare system and the patient. The high percentage of the Bangladeshi adult population at low 10-year CVD risk (<10%) highlights the potential for reduction of CVD risk through population-wide public health policy and availability of accessible preventive services. However, caution should be taken to ensure that risk stratification approaches are not used in inappropriate clinical circumstances, such as adults with highly uncontrolled hypertension with BP measurements at 160/100 mm Hg.

## CONCLUSION

Our data show that the implementation of a total risk approach compared with a single risk factor approach

will reduce the healthcare expenditure by lowering drug costs, which accounts for 60% of out-of-pocket spending in Bangladesh. This approach would be particularly beneficial in Bangladesh, a low-resource country that should prioritise the development of health policy for effective resource allocation in the public health sector. Using the total risk approach would increase service coverage and allow for the distribution of resources to target those at highest risk of experiencing a heart attack or stroke. As the majority of the Bangladeshi adult population aged ≥40 years have a low 10-year risk of CVD, strategies that target those at highest risk of CVD coupled with public health policies to reduce the population-level risk of CVD may be effective.

**Author affiliations**
[1]Non-Communicable Disease Program, World Health Organization Country Office for Bangladesh, Dhaka, Bangladesh
[2]Department of Epidemiology, Gillings School of Global Public Health, University of North Carolina at Chapel Hill, Chapel Hill, North Carolina, USA
[3]Department of Public Health, Shiga University of Medical Science, Otsu, Japan
[4]National Institute of Opthalmology, Dhaka, Bangladesh

**Acknowledgements** The study was completed by the National Institute of Ophthalmology (NIO) of Bangladesh with technical assistance from WHO Country Office for Bangladesh. The authors are grateful to the management and field team of NIO consisting of ophthalmologists, other doctors, nurses, technologists and enumerators for their hard works.

**Contributors** JYI: conceptualised the manuscript, analysed data, interpreted results critically and drafted the manuscript. MMZ: conceptualised the manuscript, designed the study, interpreted results critically, guided manuscript writing and critically reviewed it. MM, SAS and AHMEH: prepared the survey protocol, trained the field team, implemented the survey, processed the data and reviewed the manuscript. SAS is the guarantor of data.

**Funding** Financial assistance for this study was provided by the WHO Country Office for Bangladesh (WHO Reference: 2013/355662-0, Purchase Order: 200843353, Reg. File: BAN-2013-B7-TSA-0001)

**Competing interests** None declared.

**Patient and public involvement** Patients and/or the public were not involved in the design, or conduct, or reporting, or dissemination plans of this research.

**Patient consent for publication** Not required.

**Ethics approval** The authors obtained ethical approval for this study from the Institutional Review Board of the National Institute of Ophthalmology.

**Provenance and peer review** Not commissioned; externally peer reviewed.

**Data availability statement** Data are available upon reasonable request. The deidentified participant data used and/or analysed during the current study are available from the corresponding author on reasonable request. Please contact M. Mostafa Zaman at zamanm@who.int for further information and guidelines.

**ORCID iDs**
Jessica Yasmine Islam http://orcid.org/0000-0002-3690-3848
Mohammad Moniruzzaman http://orcid.org/0000-0003-2144-7111

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
