## [Reviewer comments · BMJ Open]

ARTICLE DETAILS

TITLE (PROVISIONAL)	Estimation of total cardiovascular risk using the 2019 WHO CVD prediction charts and comparison of population-level costs based on alternative drug therapy guidelines: a population-based study of adults in Bangladesh
AUTHORS	Islam, Jessica Yasmine; Zaman, MM; Moniruzzaman, Mohammad; Ara Shakoor, Shawkat; Hossain, A.H.M. Enayet

VERSION 1 – REVIEW

REVIEWER	Udaya Ranawaka Faculty of Medicine University of Kelaniya Sri Lanka
REVIEW RETURNED	02-Dec-2019

GENERAL COMMENTS	This is a secondary analysis of data from a community survey of Bangladeshi adults. The authors have (1) estimated the CVD risk using WHO/ISH risk prediction charts, and (2) compared the cost of treatment using a single risk factor vs total CVD risk approach. The article is well written, except or a few grammatical errors. Comments 1. the first statement in Article Summary: Strengths and Limitations implies that the study was a large population survey, which may be misleading. This paper is based on secondary analysis of data from a population survey on blindness (as stated in methodology).2. The WHO/ISH risk prediction tool is believed to underestimate the population CVD risk, compared to other tools such as NCEP/ATP III. The very low proportions with high CVD risk may be due to this. As a result, the estimated cost of treatment using this tool would be lower than the estimates using other risk prediction charts. The authors need to discuss this.3. Comparison with similar risk estimates in other populations, especially South Asian LMIC populations who may share the same risk factor profile and therefore similar CVD risks, should be included in the discussion.4. The authors advocate adopting the total risk approach in managing NCDs. I would like them to discuss the relative demerits of this approach too compared to the single risk approach, such as the likelihood of denying treatment to people with elevated blood pressure or cholesterol levels.5. The article repeatedly emphasizes that adopting a total risk approach would lead to lower numbers being designated as high risk and lower treatment costs. These are not novel findings. The authors are encouraged to discuss their findings with reference to data from previous publications.
---

	6. Literature review is inadequate. The authors do not discuss their findings in relation to data from previous studies. This is a major weakness in the paper.
--	---

REVIEWER	Sitanshu Sekhar Kar JIPMER
REVIEW RETURNED	07-Dec-2019

GENERAL COMMENTS	Thank you for the opportunity to review this article. This is a very relevant work from LMIC. Comments:  1. The data presented in this paper is from 2014, may not be very useful for policy making. 2. WHO has revised risk charts in 2018-19, may be a good idea to use those for generating local evidence. 3. The study setting should be described properly. What type of programme is being implemented in Bangladesh. Insurance coverage and availability of drugs in primary care, what % of NCD care is provided by private sector. 4. WHO proposes to use STEPS instrument for conducting NCD risk factor surveillance and there is a specific methodology with sample size calculation mentioned. The authors should indicate how these numbers compare with the guideline. 5. Cost calculation done by the authors appears to be very simplistic. It is a good practice to identify the health states of individuals who will require medication and what drugs to be prescribed. To do that a standard treatment guideline should be followed. The cost of hypertension care is not just the multiplication of prevalence of the condition and unit cost of drugs. Complication of hypertension with or without hypertension, presence or absence of co-morbid conditions like Diabetes will alter the calculation. To my mind these factors should be added in a model and then we should do cost calculation. Otherwise, there will be under-estimation of the cost. 6. People with high cardiovascular risk ($\geq 30\%$ and $\geq 20\%$), who are recommended four drugs [6]. The authors need to elaborate this point. 7. Referencing is not uniform in the text. [] vs superscript
--

REVIEWER	João M Pedro EPIUnit, Instituto de Saúde Pública, Universidade do Porto, Portugal
REVIEW RETURNED	10-Dec-2019

GENERAL COMMENTS	A full revision of the English is needed, but I consider that this article to be accepted a major revision should be conducted. The data presented in the table 4 are wrong (the total is less than the other values). There is not a clear description of the clinical guidelines for choosing the medical treatment. You consider the use of Simvastatine in a population without cholesterol results, the blood sugar levels are not measured in fasting - these limitations are not discussed. You don't refer to other similar studies conducted in other countries (Mozambique and Angola for example) and the results are from a study conducted in 2013, but after that you extrapolate results for a population of 2016, and we are at 2019 - please use only 2013 results and population projection.
---

VERSION 1 – AUTHOR RESPONSE

Reviewer 1 Comments:

Comment: the first statement in Article Summary: Strengths and Limitations implies that the study was a large population survey, which may be misleading. This paper is based on secondary analysis of data from a population survey on blindness (as stated in methodology).

Response: Thank you for your comment. The authors agree and we have removed this statement.

Comment: The WHO/ISH risk prediction tool is believed to underestimate the population CVD risk, compared to other tools such as NCEP/ATP III. The very low proportions with high CVD risk may be due to this. As a result, the estimated cost of treatment using this tool would be lower than the estimates using other risk prediction charts. The authors need to discuss this.

Response: The authors agree with the reviewer's point. We have included the following in the Discussion section to highlight this point: "Although implementation of a total risk approach may lead to cost-savings, the WHO prediction charts may underestimate CVD risk in certain categories of people such as those with persistent raised blood pressure $\geq 160/100$ mmHg, blood cholesterol ≥ 8 mmol/L, or those suffering from diabetes with renal disease¹. Patients who may fall in these categories should be recommended for intensive lifestyle interventions, and appropriate drug therapy. In fact, the risk models used to develop the 2019 CVD risk charts may have underestimated CVD risk due to limitations in the population data used to estimate incidences: As reported in the 2019 Lancet publication by the WHO CVD Risk Chart Working Group the data used to develop the predictions models likely included people already on cardiovascular disease prevention therapies, such as statins, which have led to an underestimate in CVD risk². In our study, we underscore the potential for underestimation of CVD risk by comparing the proportion of adults categorized as high risk ($\geq 20\%$ CVD risk) to those who would be diagnosed with hypertension (BP $\geq 140/60$) and severe hypertension (BP $\geq 160/100$). Additionally, we provided a graphical summary of common risk factors of CVD, including hypertension, hyperglycemia, and overweight and obesity. Despite our very low proportion of adults who would be recommended for treatment based on the risk prediction charts, we observed a high prevalence of these risk factors particularly in urban populations."

Comment: Comparison with similar risk estimates in other populations, especially South Asian LMIC populations who may share the same risk factor profile and therefore similar CVD risks, should be included in the discussion.

Response: Thank you. In addition to the summary we provide of prior studies conducted in Bangladesh, we have included the following to provide a description of CVD risk estimates from other South Asian countries: "In this analysis, we applied the 2019 CVD Risk Charts for South Asia, which are newly developed and now incorporate body mass index as part of the prediction chart algorithm. Our results are comparable to data from South Asia presented in the 2019 Lancet publication by the WHO CVD Risk Chart Working Group, which showed that 0% of women from both Bhutan and Nepal had a CVD risk level above 20%. Similarly, 0% of men from Bhutan, and only $<2\%$ of men from Nepal were categorized with a risk level above 20%². These data demonstrate a lower prevalence of CVD risk $\geq 20\%$ than prior reports from South Asia, which utilized the original risk prediction charts published in 2007. For example, prior data from Nepal showed that 4.3% of adults were categorized with a high ($\geq 20\%$) 10-year risk of a CVD event³. Further, analyses from a rural area of south India revealed that seventeen percent of participants had moderate to high risk (10- \rightarrow 20%) of cardiovascular events per the 2007 WHO prediction charts⁴. Finally, data collected in 2010 from Pakistan showed that 10% of adults were categorized with $\geq 20\%$ CVD risk, with 2.9% as high as

≥40%⁵. When utilizing the 2019 WHO prediction charts on the population level to measure and monitor trends in total CVD risk in recent years, policy makers should interpret the trends with caution, and potentially compare changes in trends of CVD-risk using the criteria of both the 2007 and 2019 WHO risk prediction charts.”

Comment: The authors advocate adopting the total risk approach in managing NCDs. I would like them to discuss the relative demerits of this approach too compared to the single risk approach, such as the likelihood of denying treatment to people with elevated blood pressure or cholesterol levels.

Response: Thank you for your comment. The authors agree, and we have included two graphs to illustrate this point. We include Figure 1, which summarizes the prevalence of hypertension, hyperglycemia, and overweight and obesity by rural/urban, age group and gender. This graph summarizes the prevalence of these independent conditions, which is notable despite the very low proportion of adults categorized as ≥20% CVD risk. Additionally, we include Figure 2, which demonstrates the proportion of adults who are either hypertensive or severely hypertensive by CVD risk group. We did not measure cholesterol. We have included the following in the Discussion to reflect this addition: “In our study, we underscore the potential for underestimation of CVD risk by comparing the proportion of adults categorized as high risk (≥20% CVD risk) to those who would be diagnosed with hypertension (BP ≥ 140/60) and severe hypertension (BP ≥ 160/100). Additionally, we provided a graphical summary of common risk factors of CVD, including hypertension, hyperglycemia, and overweight and obesity. Despite our very low proportion of adults who would be recommended for treatment based on the risk prediction charts, we observed a high prevalence of these risk factors particularly in urban populations.”

Comment: The article repeatedly emphasizes that adopting a total risk approach would lead to lower numbers being designated as high risk and lower treatment costs. These are not novel findings. The authors are encouraged to discuss their findings with reference to data from previous publications.

Response: The authors agree with the reviewer’s comment. We have updated the Discussion as described above and in other sections to provide more context of prior studies.

Comment: Literature review is inadequate. The authors do not discuss their findings in relation to data from previous studies. This is a major weakness in the paper.

Response: Thank you for your comment. We have updated the Discussion to include more discussion regarding estimates from South Asia and Bangladesh. Please see response to comment above.

Reviewer 2 Comments:

Thank you for the opportunity to review this article. This is a very relevant work from LMIC.

Comment: The data presented in this paper is from 2014, may not be very useful for policy making.

Response: The authors agree that the data are outdated. However, we believe that it is important to take advantage of opportunities for secondary data analyses using data collected in resource-constrained areas such as Bangladesh, as the opportunity to conduct studies on this scale are limited. Additionally, as we have applied the 2019 WHO prediction charts, we believe we have added a novel contribution to the literature. After observing that we demonstrated similar CVD risk profiles as other countries in South Asia based on the 2019 Lancet report, the authors are more confident in our results and believe that our analysis could be important. We have added the following limitation to address the reviewer’s concern: “Further, our data were collected in 2013 and may be outdated as population growth in older age groups has been observed in recent years. Our analyses should be

replicated using more recent data and future research studies should include the measurement of total cholesterol.”

Comment: WHO has revised risk charts in 2018-19, may be a good idea to use those for generating local evidence.

Response: Thank you very much for bringing this to our attention. In fact, this suggestion improves the novelty of our analysis since this is the first manuscript from Bangladesh to apply the 2019 risk charts to estimate CVD risk. We have updated our analysis and applied the 2019 WHO CVD risk prediction charts.

Comment: The study setting should be described properly. What type of programme is being implemented in Bangladesh. Insurance coverage and availability of drugs in primary care, what % of NCD care is provided by private sector.

Response: Thank you for your question. Currently, no national program or surveillance system is in place in Bangladesh to combat cardiovascular disease, excluding national guidelines and protocol management for hypertension. We have included the following in the Discussion paragraph describing out of pocket costs for CVD treatment: “Currently, Bangladesh does not offer universal health coverage or affordable health insurance plans.”

Additionally, we have included the following to provide more details on availability of CVD treatment in Bangladesh: “In 2014, an estimated 16% of health care facilities across the country (i.e. hospitals, community clinics) had the resources to diagnose, prescribe treatment for, and manage patients with CVDs⁶. District hospitals (95%) and upazila health complexes (81%), and private hospitals (77%) were more likely to provide services for cardiovascular diseases than other facilities. Only 10% of community clinics and maternal and child welfare centers, and 17% of union level facilities, which are the most accessible providers in rural areas, provided any cardiovascular services, and the services at these facilities were limited to the measurement of blood pressure or referrals⁶. Among facilities with the capacity to offer services for CVD management, about only 20% utilized established guidelines for hypertension treatment and less than one-third had essential CVD medicines readily available on-site for patients⁶. By integrating the WHO risk prediction charts into the national guidelines for management of hypertension and CVD prevention in Bangladesh, the proportion of facilities using established guidelines may increase as the charts are easy to implement, interpret, and access. Additionally, since only one-third of facilities have essential CVD medicines readily available, distributing pharmacologic treatment to those at highest risk of premature mortality due to CVD will be crucial.”

Comment: WHO proposes to use STEPS instrument for conducting NCD risk factor surveillance and there is a specific methodology with sample size calculation mentioned. The authors should indicate how these number compare with the guideline.

Response: Yes, this is correct. We utilized the same sampling methodology that was recommended for NCD risk factor surveillance similar to the approach used in 2010 for the WHO Bangladesh’s NCD Risk Factor Survey^{7 8}. We have clarified this point in the methods as follows: “We adopted a multistage, geographically clustered, probability-based sampling approach to obtain a nationally representative sample of Bangladesh, as previously described and outlined per the WHO STEPwise approach 8-11.”

The sample size was calculated using the prevalence of a very rare outcome (prevalence of blindness at 1.53%). Since NCDs measured are much more common (10-20%), we believe that the sample size was adequate for our study objective.

Comment: Cost calculation done by the authors appear to be very simplistic. It is a good practice to identify the health states of individuals who will require medication and what drugs to be prescribed. To do that a standard treatment guideline should be followed. The cost of hypertension care is not just the multiplication of prevalence of the condition and unit cost of drugs. Complication of hypertension with or without hypertension, presence or absence of co-morbid conditions like Diabetes will alter the calculation. To my mind these factors should be added in a model and then we should do cost calculation. Otherwise, there will under-estimation of the cost.

Response: Thank you for your point. The authors agree that our cost estimation is simplistic. We carried out a simple analysis to illustrate the potential cost savings of utilizing the total risk approach in a setting where the majority (60%) of the cost of CVD-related treatment is pharmacologic intervention. In order to highlight the potential use of a total risk approach beneficial for a low-resource setting, we are replicating an analysis that was done in the past in other countries conducted by Mendis et al (2011) published in the Journal of Clinical Epidemiology 5. Additionally, it is the author's hope that the results of this paper and data provided in the Appendix will potentially inform a future cost-effectiveness analysis where factors the reviewer has mentioned can be taken into account. In order to address the reviewer's concern, we have listed the following under the Strengths and Limitation section to highlight the limitations of our analysis: "The cost estimates we present are an underestimate of total costs for CVD-related treatment as the focus of this study is on cost of pharmacologic intervention only as the largest contributor to overall direct costs in Bangladesh."

Comment: People with high cardiovascular risk ($\geq 30\%$ and $\geq 20\%$), who are recommended four drugs [6]. The authors need to elaborate this point.

Response: We have updated the text to clarify as follows: "(1) people with high cardiovascular risk ($\geq 20\%$ and BP $\geq 160/100$), who are recommended for pharmacological intervention using four different types of drugs for treatment^{1 5}." This is per the WHO Prevention of cardiovascular disease: guidelines for assessment and management of total cardiovascular risk.

Comment: Referencing is not uniform in the text. [] vs superscript

Response: Thank you. We have updated to ensure consistent referencing.

Reviewer 3 Comments:

Comment: A full revision of the English is needed, but I consider that this article to be accepted a major revision should be conducted.

Response: Thank you for your comment. We have revised the manuscript for grammatical errors.

Comment: The data presented in the table 4 are wrong (the total is less then the other values).

Response: Thank you for bringing this to our attention. We have updated Table 4 to provide the correct totals.

Comment: There is not a clear description of the clinical guidelines for choosing the medical treatment. You consider the use of Simvastatine in a population without cholesterol results, the blood sugar levels are not measured in fasting - this limitations are not discussed.

Response: Thank you for your comment. We agree that these are limitations to our approach and

have included the following to the Discussion: “Although we included simvastatin in our pharmacologic cost analysis, we were unable to measure total cholesterol or confirm the medical history of participants using medical charts and relied on self-report, leading to the potential for measurement error and recall bias. Further, our data were collected in 2013 and may be outdated as population growth in older age groups has been observed in recent years. Our analyses should be replicated using more recent data and future research studies should include the measurement of total cholesterol.”

Comment: You don't refer to other similar studies conducted in other countries (Mozambique and Angola for example).

Response: Thank you for this comment. We updated the Discussion to include a summary of similar studies conducted in South Asia. Please refer to our response to Reviewer 1 above.

Comment: The results are from a study conducted in 2013, but after that you extrapolate results for a population of 2016, and we are at 2019 - please use only 2013 results and population projection.

Response: Thank you for your comment. We agree. Initially, we were interested in using the population estimates from the Bangladesh Bureau of Statistics in hopes of more accurate population data. The earliest data we could find were from 2016. However, we agree that it would be best to use the same years. We have now updated Table 4 and the Methods section to include 2013 population data made available by The World Bank.

FORMATTING AMENDMENTS (if any)

Comment: Kindly remove all your Supplementary Table in your Main Document and upload it separately under file designation "Supplementary File" in PDF Format.

Response: Thank you. We have removed the supplementary table from the Main Document and uploaded separately as requested.

1. Prevention of cardiovascular disease: pocket guidelines for assessment and management of cardiovascular disease Geneva: World Health Organization, 2007.
2. Group WCRCW. World Health Organization cardiovascular disease risk charts: revised models to estimate risk in 21 global regions. *The Lancet Global health* 2019;7(10):e1332-e45. doi: 10.1016/S2214-109X(19)30318-3
3. Khanal MK, Ahmed MS, Moniruzzaman M, et al. Total cardiovascular risk for next 10 years among rural population of Nepal using WHO/ISH risk prediction chart. *BMC research notes* 2017;10(1):120. doi: 10.1186/s13104-017-2436-9
4. Ghorpade AG, Shrivastava SR, Kar SS, et al. Estimation of the cardiovascular risk using World Health Organization/International Society of Hypertension (WHO/ISH) risk prediction charts in a rural population of South India. *Int J Health Policy Manag* 2015;4(8):531-6. doi: 10.15171/ijhpm.2015.88
5. Mendis S, Lindholm LH, Anderson SG, et al. Total cardiovascular risk approach to improve efficiency of cardiovascular prevention in resource constrain settings. *Journal of clinical epidemiology* 2011;64(12):1451-62. doi: 10.1016/j.jclinepi.2011.02.001
6. NIPORT. Bangladesh Health Facility Survey 2014 Dhaka, Bangladesh: National Institute of Population Research and Training (NIPORT), Associates for Community and Population Research (ACPR), and ICF International. ; 2016 [Available from: <https://dhsprogram.com/pubs/pdf/SPA23/SPA23.pdf> accessed 03/15 2019.
7. Moniruzzaman M, Mostafa Zaman M, Islalm MS, et al. Physical activity levels in Bangladeshi adults: results from STEPS survey 2010. *Public health* 2016;137:131-8. doi:

10.1016/j.puhe.2016.02.028

8. Zaman MM, Rahman MM, Rahman MR, et al. Prevalence of risk factors for non-communicable diseases in Bangladesh: Results from STEPS survey 2010. Indian journal of public health 2016;60(1):17-25. doi: 10.4103/0019-557X.177290

9. Rahman M, Zaman MM, Islam JY, et al. Prevalence, treatment patterns, and risk factors of hypertension and pre-hypertension among Bangladeshi adults. Journal of human hypertension 2018;32(5):334-48. doi: 10.1038/s41371-017-0018-x

10. Karim MN, Zaman MM, Rahman MM, et al. Sociodemographic Determinants of Low Fruit and Vegetable Consumption Among Bangladeshi Adults: Results From WHO-STEPs Survey 2010. Asia-Pacific journal of public health / Asia-Pacific Academic Consortium for Public Health 2017;29(3):189-98. doi: 10.1177/1010539517699059

11. Surveillance of risk factors for noncommunicable disease: the WHO STEPwise approach. Geneva: World Health Organization, 2001.

VERSION 2 – REVIEW

REVIEWER	Udaya Ranawaka University of Kelaniya Sri Lanka
REVIEW RETURNED	17-Feb-2020

GENERAL COMMENTS	1. I am not quite sure of the validity of using the 2019 charts for a retrospective analysis of data from a study done in 2013.2. The authors have now added references from some of the South Asian countries, but there are several publications on cardiovascular risk estimates from other South Asian countries not covered in the discussion. It would be better if these too are included to make the discussion more comprehensive in its coverage of the CVD risk assessments in the region. I note that other reviewers too have commented on the need to include references from other countries.3. I do not think the authors have adequately addressed the following comment - "The authors advocate adopting the total risk approach in managing NCDs. I would like them to discuss the relative demerits of this approach too compared to the single risk approach, such as the likelihood of denying treatment to people with elevated blood pressure or cholesterol levels."
---

REVIEWER	João M Pedro EPIUnit, Instituto de Saúde Pública, Universidade do Porto, Porto, Portugal
REVIEW RETURNED	21-Feb-2020

GENERAL COMMENTS	The authors responded to all questions and suggestions raised, except on the limitations section, but this do not prevent the paper to be published.
--

VERSION 2 – AUTHOR RESPONSE

Reviewer 1 Comments:

Comment: I am not quite sure of the validity of using the 2019 charts for a retrospective analysis of data from a study done in 2013.

Response: Thank you for your comment. We used the 2019 charts based on the recommendation of Reviewer 2. The original charts were published in 2007. Unlike the original 2007 charts which were derived using data only from high-income countries, the 2019 updated charts are based on adapted risk prediction models that now includes data from low- and middle-income countries[1]. Additionally, to develop the risk prediction models, the WHO CVD working group used country specific WHO- STEPS surveys which were collected from calendar years 2002-2017. After the prediction models were developed, the working group conducted external validation and the data used for this process were from studies conducted as early as the 1970s (1974-2013).

If necessary, for further details please refer to Tables 1.2 and 1.4 in Appendix 1 of the 2019 paper published in Lancet titled "World Health Organization cardiovascular disease risk charts: revised models to estimate risk in 21 global regions."

Based on the methods and data used by the WHO to develop the 2019 prediction charts, we believe using the 2019 charts is suitable for our cohort.

Comment: The authors have now added references from some of the South Asian countries, but there are several publications on cardiovascular risk estimates from other South Asian countries not covered in the discussion. It would be better if these too are included to make the discussion more comprehensive in its coverage of the CVD risk assessments in the region. I note that other reviewers too have commented on the need to include references from other countries.

Response: Thank you for your comment. The authors attempted to include studies that are categorized as "South Asia," according to the 2019 WHO CVD Risk Charts (Bangladesh, Bhutan, India, Nepal, and Pakistan). The 2019 charts now recommend a separate chart for "South East Asia," which includes countries such as Sri Lanka and Myanmar that traditionally are categorized as "South Asia." However, the authors agree that a more comprehensive literature review could be included. The following has been either updated or added to the Discussion. The new text has been underlined. "Our results are comparable to regional data presented in the 2019 Lancet publication by the WHO CVD Risk Chart Working Group, which show that 0% of women from Bhutan, Sri Lanka, and Nepal have a CVD risk level above 20%. Similarly, 0% of men from Bhutan, and only <2% of men from both Sri Lanka and Nepal were categorized with a risk level above 20%[1]. These data demonstrate a lower prevalence of CVD risk $\geq 20\%$ than prior reports from South Asia, which utilized the 2007 risk prediction charts. For example, prior data from Nepal [2] and Sri Lanka[3] showed that 4.3% and 8.2% of adults respectively, were categorized with a high ($\geq 20\%$) 10-year risk of a CVD event. Further, analyses from a rural area of South India revealed that seventeen percent of participants had moderate to high risk (10->20%) of cardiovascular events[4]. Finally, data collected in 2010 from Pakistan showed that 10% of adults were categorized with $\geq 20\%$ CVD risk, with 2.9% as high as $\geq 40\%$ [5]. When utilizing the 2019 WHO prediction charts on the population level to measure and monitor trends in total CVD risk in recent years, policy makers in low- and middle-income countries should interpret the trends with caution, and assess changes in trends of CVD-risk over time using both the 2007 and 2019 WHO risk prediction charts for comparison.

The WHO CVD Risk Charts were developed for use in low- and middle-income countries and are now more suitable for use in these settings due to the inclusion of data from low-resource regions in the risk prediction model development. While we present the first country-specific analysis using the 2019 risk charts, several prior studies in low- and middle-income countries have been conducted using the 2007 risk charts. In other countries in Asia, we observe the prevalence of "high CVD risk" ($\geq 20\%$) of 6.0%, 2.3% and 1.3% in Mongolia, Malaysia and Cambodia, respectively[6]. Mendis et al reported the 10 year CVD risk of seven countries and the majority of these countries reported low CVD risk among its adult populations [China (96.1%) and Sri Lanka (94.9%), (Iran (93.9%), Cuba (89.7%), Nigeria

(86.0%), Georgia (83.1%), and Pakistan (79.2%)[5]. Studies conducted in urban areas of low- and middle-income countries show varying prevalence of high CVD risk ($\geq 20\%$): for example, one study from Malaysia shows 20.5% of adults were high-risk of CVD[7], whereas studies from urban Kenya[8] and Sri Lanka[3] reports less than 10% of their population had high CVD risk. Specifically in Bangladesh, utilizing the 2007 prediction charts, three prior studies have reported absolute CVD risk among the adult population [9-11]. Fatema et al. found that 10% of rural Bangladeshi adults were at high risk ($\geq 20\%$) of a CVD event within the next ten years, and half of these adults fell in the very high-risk category ($\geq 30\%$). In another rural Bangladeshi population, the proportion of participants at a high-risk ($\geq 20\%$) of a CVD event 2.1%[10]. Finally, in an urban Bangladeshi population of 150 adults, 3.4% had high CVD risk ($\geq 20\%$), which is lower than expected as the population was urban[11].

Comment: I do not think the authors have adequately addressed the following comment - "The authors advocate adopting the total risk approach in managing NCDs. I would like them to discuss the relative demerits of this approach too compared to the single risk approach, such as the likelihood of denying treatment to people with elevated blood pressure or cholesterol levels."

Response: Thank you for your comment. We have included the following in the Discussion: "Although implementation of a total risk approach may lead to cost-savings, there are limitations to implementing the 2019 CVD Risk Prediction Charts. When compared to the single risk factor approach, the WHO charts categorizes fewer individuals as high-risk and may delay the receipt of necessary life-saving treatment. For example, the prediction charts may underestimate CVD risk in certain categories of people such as those with persistent raised blood pressure $\geq 160/100$ mmHg, blood cholesterol ≥ 8 mmol/L, or those suffering from diabetes with renal disease[12]. Patients who may fall in these categories should be recommended for intensive lifestyle interventions, and appropriate drug therapy, however, the CVD prediction charts will erroneously deny treatment to these potentially high-risk adults. In fact, the risk models used to develop the 2019 CVD risk charts may have underestimated CVD risk due to limitations in the population data used to estimate incidences: Data used to develop the predictions models likely included people already on cardiovascular disease prevention therapies, such as statins, which have led to an underestimate in CVD risk[1]. In our study, we underscore the potential for underestimation of CVD risk by comparing the proportion of adults categorized as high risk ($\geq 20\%$ CVD risk) to those who would be diagnosed with hypertension (BP $\geq 140/60$) and severe hypertension (BP $\geq 160/100$). Additionally, we provided a graphical summary of common risk factors of CVD, including hypertension, hyperglycemia, and overweight and obesity. Despite our very low proportion of adults who would be recommended for treatment based on the risk prediction charts, we observed a high prevalence of these risk factors particularly in urban populations."

Reviewer 3 Comments:

Comment: The authors responded to all questions and suggestions raised, except on the limitations section, but this do not prevent the paper to be published.

Response: Thank you for your comment and pointing out this missed suggestion. We agree and have included the following limitation in the Discussion section. Please recall that both diabetes and cholesterol levels was not included in the 2019 non-laboratory CVD risk charts as described in the Methods : "Additionally, we were unable to categorize participants as diabetic as we did not obtain fasting blood glucose and were only able to categorize adults as hyperglycemic in our descriptive analysis as we measured random blood glucose."

1. Group, W.C.R.C.W., World Health Organization cardiovascular disease risk charts: revised models to estimate risk in 21 global regions. *Lancet Glob Health*, 2019. 7(10): p. e1332-e1345.

2. Khanal, M.K., et al., Total cardiovascular risk for next 10 years among rural population of Nepal using WHO/ISH risk prediction chart. *BMC Res Notes*, 2017. 10(1): p. 120.

3. Ranawaka, U.K., et al., Risk estimates of cardiovascular diseases in a Sri Lankan community. Ceylon Med J, 2016. 61(1): p. 11-7.
4. Ghorpade, A.G., et al., Estimation of the cardiovascular risk using World Health Organization/International Society of Hypertension (WHO/ISH) risk prediction charts in a rural population of South India. Int J Health Policy Manag, 2015. 4(8): p. 531-6.
5. Mendis, S., et al., Total cardiovascular risk approach to improve efficiency of cardiovascular prevention in resource constrain settings. J Clin Epidemiol, 2011. 64(12): p. 1451-62.
6. Otgontuya, D., et al., Assessment of total cardiovascular risk using WHO/ISH risk prediction charts in three low and middle income countries in Asia. BMC Public Health, 2013. 13: p. 539.
7. Su, T.T., et al., Prediction of cardiovascular disease risk among low-income urban dwellers in metropolitan Kuala Lumpur, Malaysia. Biomed Res Int, 2015. 2015: p. 516984.
8. Vusirikala, A., et al., Assessment of cardiovascular risk in a slum population in Kenya: use of World Health Organisation/International Society of Hypertension (WHO/ISH) risk prediction charts - secondary analyses of a household survey. BMJ Open, 2019. 9(9): p. e029304.
9. Fatema, K., et al., Application of two versions of the WHO/international society of hypertension absolute cardiovascular risk assessment tools in a rural Bangladeshi population. BMJ Open, 2015. 5(10): p. e008140.
10. Cravedi, P., et al., Preventing renal and cardiovascular risk by renal function assessment: insights from a cross-sectional study in low-income countries and the USA. BMJ Open, 2012. 2(5).
11. Ahmed, M.S.A.M., et al., Cardiovascular Risk Assessment Among Urban Population of Bangladesh Using WHO/ISH Risk Prediction Chart. International Journal of Epidemiology, 2015. 44(suppl_1): p. i202-i202.
12. Prevention of cardiovascular disease: pocket guidelines for assessment and management of cardiovascular disease 2007, World Health Organization: Geneva.

VERSION 3 – REVIEW

REVIEWER	Udaya Ranawaka University of Kelaniya Sri Lanka
REVIEW RETURNED	22-Apr-2020
GENERAL COMMENTS	The authors have adequately addressed the queries and comments from the previous review. I am happy for the article to be accepted for publication.